# Mental disorders following electrical injuries— A register-based, matched cohort study

**Karin Biering** *, **Jesper Medom Vestergaard, Anette Kærgaard, Ole Carstensen, Kent J. Nielsen**

Department of Occupational Medicine–University Research Clinic, Danish Ramazzini Centre, Regional Hospital West Jutland, Herning, Denmark

* karbie@rm.dk

## Abstract

### Introduction

Electrical injuries happen every day in homes and workplaces. Not only may these injuries cause physical damage and disability, they may also cause mental disorders. The aim of this study was to investigate if persons with an electrical injury suffer from mental disorders in the following years.

### Material and methods

In a prospective matched cohort design, we identified 14.112 electrical injuries in two Danish registries and matched these with persons with dislocation/sprain injuries or eye injuries, respectively, as well as with persons from the workforce from the same occupation, using year of injury, sex and age as matching variables. We identified possible outcomes in terms of mental diagnoses in the Danish National Patient registry, based on literature, including reviews, original studies and case-reports as well as experiences from clinical praxis. The associations were analyzed using conditional cox- and logistic regression.

### Results

We found that the following of the examined outcomes were associated with exposure to an electrical injury compared to the matched controls. Some of the outcomes showed the strongest associations shortly after the injury, namely 'mental disorders due to known physiological condition', 'anxiety and adjustment disorders', and especially the 'Post Traumatic Stress Disorder (PTSD)' subgroup. The same pattern was seen for 'Depression' although the associations were weaker. Other conditions took time to develop ('Somatoform disorders'), or were only present in the time to event analysis ('other non-psychotic mental disorders' and 'sleep disorders'). The findings were consistent in all three matches, with the highest risk estimates in the occupation match.

### Conclusion

Electrical injuries may result in mental disorders, both acute and several years after. However, the absolute risk is limited as most of the outcomes are rare.

**Data Availability Statement:** The data underlying the results presented in the study are owned by Statistics Denmark. Only Danish research environments are granted authorization to

Statistics Denmark. Foreign researchers can, however, get access to micro data through an affiliation to a Danish authorized environment. The data are available by application to Statistics Denmark (https://www.dst.dk/en/TilSalg/skraeddersyede-loesninger). Additional inquiries may be sent to dst@dst.dk.

**Funding:** KB, JMV, OC, AK and KN received funding from the Danish Working Environment Research Fund, grant number 22-2017-09. https://amff.dk/about-the-fund/ The funders had no role in study design, data collection and analysis, decision to publish, or preparation of the manuscript.

## Introduction

Electrical injuries occur in homes and at workplaces, despite increased technical safety and rules and regulations for handling electricity. Exposure to electrical current may result in a short pain and scare, but may also cause an injury with physiological damage such as tissue damage, burns or even heart failure or death [1]. The extent of the physical damage is related to voltage, point of current entry and pathway, duration and if the person was involuntarily stuck to the power source (no-let-go) [2]. The incidence rate of electrical injuries in Denmark is difficult to estimate as non-fatal electrical injuries are not systematically reported. This is also the case in other countries. Mortality from electrical injuries is rare in Denmark, with less than two cases on average per year from 1996 to 2005 [3].

Besides the physiological damages, electrical injuries have been reported to cause psychological problems, such as post-traumatic stress disorder(PTSD) [4–7], depression [5–9], anxiety [6–8, 10, 11], sleep disorders [5–7, 10, 11], cognitive problems [6, 8, 11, 12] and sexual dysfunction [9, 11]. An American retrospective study evaluated psychiatric examinations of 73 referred electrical injured patients and found that having experienced "no-let-go" or having lost or impaired consciousness was associated with psychiatric diagnosis [13]. The same research team repeated the study with 86 new post-acute electrical injured patients referred for treatment and supplemented with neuropsychological evaluations. They found that 78% of the subjects had one or more psychiatric diagnoses and that those with diagnosis also had poorer cognitive performance [14]. Follow-up time varied tremendously in both studies, from acute up to nine years [13, 14].

A French register-study on electrical injuries report that neuropsychiatric sequelae is the second most common type of sequelae after those directly related to burns [4]. In a small long-term follow-up study from the US, the neuropsychological problems were persistent more than a decade after the initial electrical injury [15]. In a follow-up study of 40 electrical injured patients, early emotional sequelae was found to predict poor outcomes 4 years after the injury in terms of adjustment to injury, psychological distress and return to work [16]. A larger multicenter study from Canada with 114 patients, who were interviewed shortly after the injury and again one year later, found that several neuropsychological complaints were persistent over time or even increased [17]. However, a recent Swedish study found no long-term cognitive dysfunctions [12].

Andrews et al. suggest that these neuropsychological symptoms relate to a specific syndrome related to electrical injuries, in line with post-commotional syndrome [18] and expand this in a recent review with suggestions that several types of cognitive problems are present following an electrical injury [19]. A study in an outpatient burn unit found that electrical injured patients are often referred to specialty consultation in various specialties, with psychology as the most frequent, and often the consultations does not lead to relevant findings [20].

A review of the literature from 2013 concluded that neuropsychological sequelae are common, including behavioral and cognitive changes, irritability, frustration as well as depression and PTSD. The sequelae were similar to symptoms resulting from traumatic brain injury [21]. A common problem in the literature was that the pre-injury mental health was not taken into account, since the studies use retrospective methods or are case studies. Another problem is that persons with electrical injuries are rarely compared to any control group. We identified one exception from this; a study from 1998 that compared electrical injury patients with electricians without history of electrical injury and found that the injured group had more physical, cognitive and emotional complaints [9]. We were not able to find any previous studies that had dementia as outcome, as previous studies focused on cognitive symptoms in terms of e.g. memory loss and concentration problems.

Swedish researchers recently interviewed 23 male electricians who reported different problems after an electrical injury, and found that the respondents reported both acute emotional reactions as well as long-term consequences in terms of anxiety, fear of performing the same task, feeling of guilt or anger and cognitive impairments. The study included only electricians with complaints, some even several years after the injury [22].

The existing evidence on mental and psychological problems following electrical injuries is based on either retrospective studies, studies without comparison groups or small case studies often conducted several years after the injury. We aim to improve this to a study using diagnoses from a national register and not self-reported symptoms as outcomes in a matched design.

The aim of this study was to investigate if persons with an electrical injury suffer from mental disorders in the following years in a prospective matched cohort study. This is based on the hypothesis, that electrical injuries more often result in mental health problems compared to other injuries.

## Material and methods

### Material

This study was a matched cohort study based on injuries registered in two population-based registers: The Danish National Patient Registry (DNPR) and the registry of reported occupational injuries from The Danish Working Environment Authority (DWEA). Furthermore, it included data from other population based registers in Statistics Denmark, described in detail in the following.

The DNPR covers all hospital contacts in Denmark, including information regarding injuries, diagnoses and procedures for both in- and outpatients and emergency department visits [23, 24]. The DWEA register contains occupational injuries reported by employers, employees, unions and health staff. In Denmark, it is mandatory for employers to report any work injury causing sick leave at least the day following the day of the injury. The DWEA register is aimed for compensation claims, but additionally, the reporting system is designed for surveillance of occupational injuries [25].

### Methods

The study covered registered Danish electrical injuries in either the DNPR (from 1994 to -2016) or DWEA (from 2005 to -2016). We combined the two registers to identify as many cases with an electrical injury as possible. We included electrical injuries starting from 1996 and up to 2014 in the study to allow for at least two years of time clear of the outcomes of interest before the injury and at least two years for the outcomes to occur after the injury. If identified in DWEA the injury was an occupational injury, while this was not necessarily the case if the injury was identified in DNPR.

**Participants.** In DNPR persons with hospital visits due to electrical injuries were identified by selecting contacts coded with the ICD10 classifications DT754 (electrocution), as well as the Danish mechanisms of injury classifications EUHA10 (Release of electrical energy), EUYD4 (Electrical installations / systems), EUWA10 (Self-harm action with electrical energy) or EUYZ0020 (Electric current). The DT754 code was used in the whole study period, while the injury codes (EU*) were used only from 2000 onwards where a separate injury register was established. Both injured persons admitted to the hospital and in outpatient clinics were included. In DWEA persons with electrical injuries were identified using information regarding the exposure either "Acute/short exposure to welding arc or electric arc." or "Acute/short exposure to electricity or reception of electric charge in the body".

To avoid that the same accident, registered with a small deviation of dates were analysed as two accidents only the registration in DWEA was used if an accident was registered in both registers (+/- 7 days). Only the first injury for each person was used, regardless of origin.

Injury records from the DNPR and DWEA were linked to Statistics Denmark, using a unique personal identification number and injury date/year. Each Danish citizen as well as registered migrant workers holds this number that provide the possibility to link each person across different registries [26].

Statistics Denmark is the central authority on Danish registries and statistics. In this study we used the following registries: the population register (to derive sex and age at time of matching) [27], the Register-based Labour Force Statistics(RAS) register (to determine if the participant was part of the workforce at the time of the injury and to identify match persons from the workforce) [28], the migration register (to derive date for possible migration) [29] and the register of deaths (to derive date for possible death) [30].

For sensitivity analysis, we derived information about length of hospitalization from DNPR. We furthermore investigated whether the persons were given a concussion diagnosis (DS06.0) at the time of the injury to be able to take into consideration if the injury had other consequences i.e. if the persons had fallen, as found in a prospective study from Iran where electrical injuries were also related to other physical injuries [31]. In that case, the outcomes could be related to post-commotional syndrome. Several authors describe traumatic brain injury in relation to electrical injury [32, 33]. Hospitalization, including time in the emergency department, was calculated for all hospital admissions. We used this as a proxy for severity, under the assumption that the most severe injuries would result in the longest hospital stay. Not all injuries from DWEA could be assigned a length of hospitalization, as we could not identify any hospital contact at the time of the injury.

**Matching.** Each person was matched in three different ways with persons from the same data source (DNPR or DWEA). We chose to make three different matches, as the perfect match-person was difficult to define. It should be a common injury, to make sure that is was possible to find a sufficient number of match persons, and it should not be related to the outcomes of interest. We decided on match persons with a dislocation/sprain injury and with eye injuries. Furthermore, we made an additional match using match persons from the workforce with the same occupation as described below. Persons for whom at least one match-person could not be identified were excluded. Match persons could be used more than once, and persons with an electrical injury could be used as controls before the electrical injury.

*Match 1: Injury-match dislocation/sprain.* Electrical injured persons were matched with up to ten other persons with a dislocation/sprain (DS93 in DNPR and sprains in DWEA).

*Match 2: Injury-match eye.* Electrical injured persons were matched with up to ten other persons with an eye injury (DT15 in DNPR). We did not identify eye injuries in DWEA due to lack of information regarding eye injuries.

The matching variables were sex, age and year of injury. The latter was included as registration practice for electrical injuries differed over time, and we cannot rule out that this could be the case for some of the outcomes as well, although ICD-10 codes are always used as opposed to codes describing injuries. For all matches the match-persons were randomly chosen, if more than ten were available per person. Due to this random process, it was possible for the same person to act as a match-person for more than one electrical injury, but only in that particular year, as only the first event was used. If the exact age was not possible to match, the algorithm identified the closest persons in age within the same 5-year age group, but with the same sex and injury-year.

*Match 3: Occupation match.* Electrical injury persons were matched with up to ten other persons from the working population, with the same occupational group, sex and age in the

year of the electrical injury. The injured person and the match-persons were working at the time of the match, however the electrical injury registered in DNPR could have happened outside work. Match-persons were given a fictive injury-date, based on their match-person's injury, to be able to identify outcomes before and after a specific point in time. The purpose of this match was to take into account that persons with certain occupations could be in higher risk of the outcomes due to socioeconomic factors or other occupational exposure rather than exposure to electrical injury.

**Outcomes.**   We selected a wide range of possible outcomes a priori, based on literature, including reviews, original studies and case-reports as well as experiences from clinical praxis at our department of occupational medicine. These outcomes were examined one by one. The outcomes with ICD-10 diagnosis were: Alzheimer's (DF00, DG30), Dementia (DF01, DF02, DF03, DF04), Mental disorders due to known physiological condition (DF06, DF07, DF09), Depression (DF32, DF33), Anxiety and adjustment disorders (DF40, DF41, DF42, DF43, DF44), Somatoform disorders (DF45), Other non-psychotic mental disorders (DF48), Sleep disorders (DF51) and Sexual dysfunction (DF52). Furthermore, we analyzed PTSD (DF43.1) as a subgroup. These ICD-10 diagnoses did not match all outcomes mentioned in the previous literature, where some outcome were wider, i.e. psychiatric diagnosis or narrower, based on self-reported symptoms i.e. memory loss or irritability, that is symptoms not covered by a single ICD-10 code, but included in others.

We excluded persons (both electrical-injury persons and match-persons), if they were registered with the outcome of interest before the matching. This was done for each outcome separately, to keep the persons in the dataset to examine other outcomes. If electrical injured persons were excluded, all their matching controls were also excluded, while match-persons were excluded one by one, keeping the remaining match-persons and the exposed person in the dataset. This implies that for each analysis of a specific outcome, the study sample was different.

In the occupation match, persons with an injury registered in DWEA were all defined as part of the working population, as their injury had occurred while working. However, not all of them were defined as part of the workforce in Statistics Denmark, most likely because they had experienced the electrical injury in at part-time job (students, interns or retired persons). This implies that 175 persons with an injury registered in DWEA could not be matched in the occupation match but only in the injury match with dislocation/sprain injuries as controls.

Occupation was derived from the RAS register at Statistics Denmark using DISCO-codes. DISCO is the official Danish version of International Standard Classification of Occupations, ISCO, prepared by the International Labor Organization(ILO) [34]. For matching groups, we used the second level (two-digit) in the hierarchy. Current working status was also derived from the RAS register, to define persons from the working population.

Both electrical injured persons and controls, who emigrated or died during follow-up, were excluded from that date, as follow-up in DNPR then was impossible.

Mandatory registration of accidents in DNPR started in 2000. Before that, the accident code (DT754) was sometimes used, but not necessarily if the main problem following the accident was something else, like a burn or unconsciousness.

In DWEA registration of accidents has been taken place since 1916, but the data available in Statistics Denmark was from 2004 to 2017. Underreporting is a well-known issue in DWEA [35].

**Statistical methods.**   We compared the two matching groups using conditional logistic regression where each match group consisted of one injured person and up to 10 match-persons, depending on availability, and exclusions for each specific outcome. We also conducted a Cox regression, to examine the outcomes in a time to event setting, to examine if any off the outcomes would occur earlier for persons who had experienced an electrical injury. Test of Schoenfeld's residuals were used to confirm proportional hazard.

The injuries were a combination of occupational injuries and injuries in other settings. From DNPR we did not have information about the setting, but we tried to accommodate this by an additional analysis in the dataset matched on injury including only persons in the workforce. Furthermore, we made a subgroup analysis in match 1 and 2, including only persons in the workforce at the time of the injury.

As a sensitivity analysis, we excluded 43 persons (10 with electrical injury and 33 dislocation/sprain controls) if they were registered with the diagnosis concussion(DS06.0) at the same time as the accident and repeated the analysis for the three outcomes that could be related to concussion; Alzheimer's, Dementia and Mental disorders due to known physiological condition. We also tried to exclude persons with a hospitalization shorter than 1 day to study if more severe accidents would reveal stronger associations. Finally, we stratified on gender, to identify any differences. All sensitivity analyses were performed in match 1.

All procedures performed in this study were in accordance with Danish ethical standards and with the Helsinki Declaration. The Regional Data Protection Agency has approved the study (reference number 1-16-02-113-18. According to Danish law, register-based studies only need approval from the ethics committee if the data include human biological material (§ 14 in 'Promulgation of the Act on the ethical treatment of health science research projects and health data science research projects' available in Danish from www.retsinformation.dk/eli/lta/2020/1338). All data were stored and treated in a secure, protected server at Statistics Denmark. Results from Statistic Denmark are only accessible for the researcher at an aggregated level, not at the individual patient level.

## Results

We identified 20,155 electrical injuries in DNPR and 1,810 in DWEA (Fig 1 and Table 1, previously published in Biering et al. [39] available from: https://oem.bmj.com/content/78/1/54). After exclusion of persons < 18 years, persons without a valid identification-number and persons who died within the first 2 days after the accident, there was an overlap of 817 persons from the two registers. Invalid identification-numbers could originate from tourists or migrant workers with a temporary number as well as mistyping in DWEA. When removing the overlap and keeping only the first electrical injury for each person we had 13,317 from DNPR and 795 from DWEA for the injury matches. For the occupation match we furthermore excluded 2,646 persons not in the workforce. We gave priority to DWEA registrations, if there was a double registration, and thus we had 10,764 injuries from DNPR and 702 injuries from DWEA available for the occupation match. A match with 10 match persons was possible for almost all electrical injuries.

The majority of the injuries happened to men, especially in DWEA, and younger persons were overrepresented, most evident in DNPR. The occupations with most injures were craft workers, but also service workers/sales workers were overrepresented, even when comparing to the distribution of occupations in Denmark. The length of hospitalization was in most cases less than one day. Over the study period, we noticed that the number of electrical injuries in DNPR increased, while the opposite is the case in DWEA (Biering et al. [39]). For each outcome, we excluded those who had experienced the outcome before the injury. The numbers of these exclusions are provided in Table 1. Frequencies of persons with the outcomes during 5 years follow up and during full follow-up are also provided in Table 1. The numbers reflect that some of the outcomes are very rare, which affects the precision of the estimates in the following tables.

We provide the risk estimates in separate tables for each match (Table 2). Due to some of the outcomes being rare, the estimates are not always possible to obtain or have wide confidence intervals. We provide the results for these underpowered analyses, but encourage cautious interpretation.

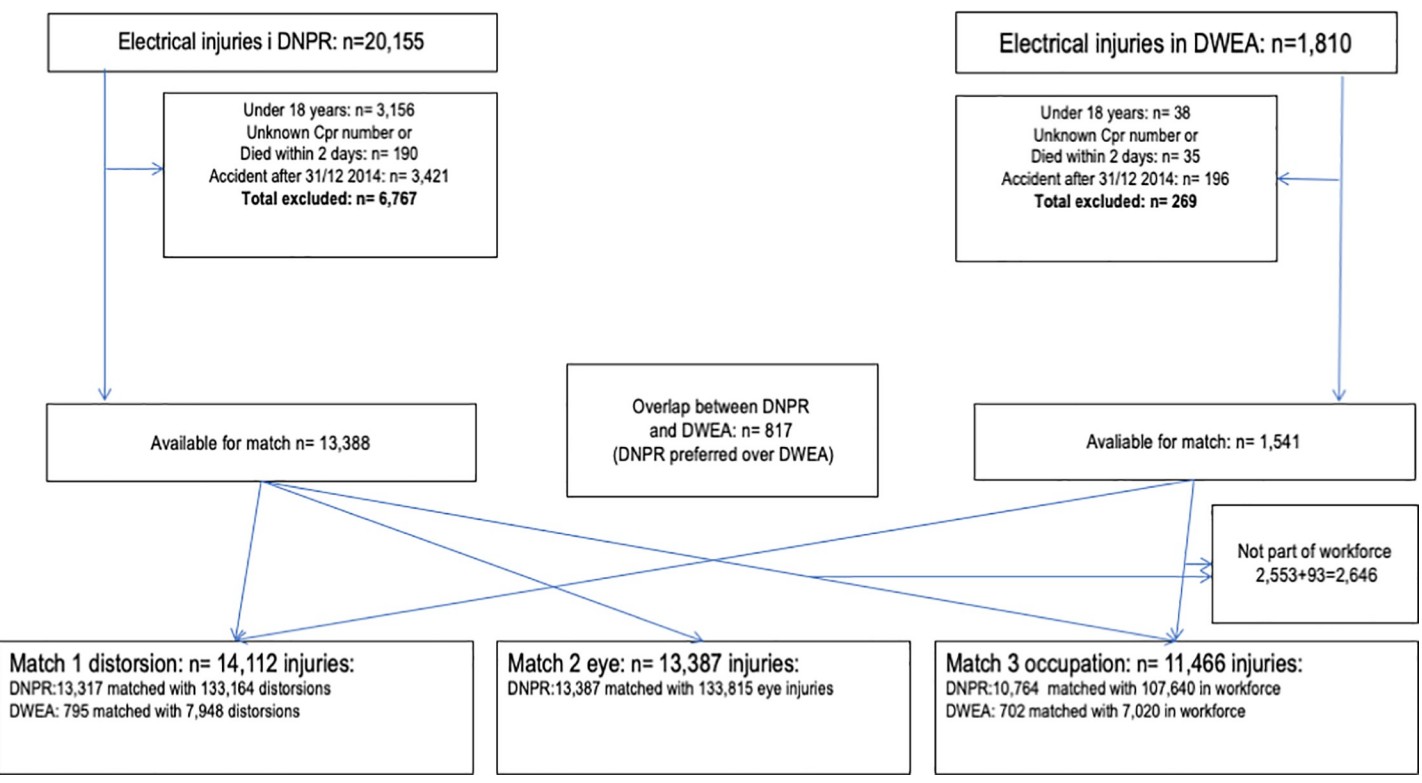

**Fig 1.**

We did not find increased risk of Alzheimer's or Dementia in any of the three matches, however, there was an exception for the analysis of only the working population in match 1 and 2, that showed increased risk of Alzheimer's. In match 3, there were also a hazard ratio of 2.13, however, not statistically significant (Table 2).

We found increased risk of 'mental disorders due to known physiological condition' in all three matches and furthermore found that the risk was highest shortly after the injury in match 1, but highest after around 2 years in the other two matches. All estimates had wide confidence intervals, especially shortly after the injury.

We found only limited associations between electrical injuries and depression. Again, there was a tendency that the estimates were highest shortly after the injury and slightly higher for the working population compared to the full population.

In all three matches, there was an increased risk of 'anxiety and adjustment disorders', especially 'Post Traumatic Stress Disorder (PTSD)' that showed extremely high estimates shortly after the injury in match 1 and 2. In match 3, there were too few cases to estimate risk at 6 and 12 months, but after 2 years the risk was still extremely high.

Risk of 'Somatoform disorders' seemed to increase over time in the two first matches, while the same pattern was difficult to identify in match 3, although the associations also were high (OR and HR above 2.3 in all time points and in the time to event analysis).

Diagnosis of 'other non-psychotic mental disorders' seemed to be associated in the time to event analysis in the two first matches, but not in match 3, but also here the confidence intervals were wide.

Finally, we found 'sleep disorders' associated with exposure to electrical injuries, but the risk estimates were in general smaller and not consistent over time. The risk was higher in

**Table 1. Exclusions and outcomes for each of the three matches.**

| | Match 1: Dislocations/sprains (14.112 injuries) | | | Match 2: Eye (13.387 injuries) | | | Match 3: Occupation (11.466 injuries) | | |
|---|---|---|---|---|---|---|---|---|---|
| | Excluded due to previous outcome | Outcomes during 5 years (n/%) | Outcomes during full follow-up | Excluded due to previous outcome | Outcomes during 5 years (n/%) | Outcomes during full follow-up | Excluded due to previous outcome | Outcomes during 5 years (n/%) | Outcomes during full follow-up |
| **Outcome** | n | n | n | n | n | n | n | n | n |
| Alzheimer's dementia | <5 | <5 | 9 | <5 | <5 | 9 | 0 | 0 | 6 |
| Dementia | <5 | <5 | 19 | <5 | <5 | 18 | <5 | 0 | 7 |
| Mental disorders due to known physiological condition | 11 | 16 | 35 | 12 | 15 | 34 | 5 | 12 | 23 |
| Depression | 71 | 71 | 138 | 70 | 67 | 132 | 38 | 42 | 95 |
| Anxiety and adjustment disorders | 145 | 135 | 242 | 144 | 130 | 236 | 89 | 85 | 171 |
| Post Traumatic Stress Disorder (PTSD) | <5 | 17 | 26 | <5 | 15 | 24 | <5 | 11 | 16 |
| Somatoform disorders | 30 | 15 | 23 | 27 | 14 | 22 | 17 | 8 | 14 |
| Other non-psychotic mental disorders | 8 | <5 | 7 | 7 | <5 | 7 | 5 | <5 | <5 |
| Sleep disorders | 5 | <5 | 10 | 5 | <5 | 10 | <5 | <5 | 9 |
| Sexual dysfunction | 21 | 8 | 20 | 21 | 7 | 18 | 19 | 7 | 18 |

Due to Statistic Denmarks rules of reporting data, cells with less than five persons are given the same value: <5.

match 3 and in the subgroups of the working population. We did not identify increased risk of 'sexual dysfunction'.

In general, all estimates were highest in match 3, where the match persons were controls without injury, but with the same occupation. In match 1 and 2, the risk estimates were often higher when restricting the analysis to the working population only.

The sensitivity analyses performed on match 1 (Table 3) showed that when we restricted the analysis for hospitalizations one day or longer, most of the risk estimates were slightly attenuated. Excluding persons who were given a diagnosis for concussion during the same stay as the electrical injury did not change the estimates. When we stratified the analyses on gender, we found that the risk of 'mental disorders due to known physiological condition' was higher for women compared to men, while the risk for 'somatoform disorders' was higher for men compared to women.

## Discussion

This is, to our knowledge, the first study using a matched design based om population based registers to investigate the association between exposure to electrical injury and mental disorders, and thus it is difficult to compare directly to previous studies. We studied 14,112 electrical injuries identified over a period of 19 years. We found that several of the examined outcomes were associated with exposure to an electrical injury compared to the matched controls. Some of the outcomes showed the strongest associations shortly after the injury, namely

**Table 2. Associations between electrical injuries and outcomes over the whole study and in intervals (electrical injuries matched with dislocation/sprain injured, eye injured and occupation controls).**

| | | Time to event | Time to event Workforce only | 6 months | 12 months | 2 years | 3 years | 4 years | 5 years |
|---|---|---|---|---|---|---|---|---|---|
| Outcome | Match | HR | HR | OR | OR | OR | OR | OR | OR |
| Alzheimer's dementia | 1 Dislocation/sprain | 1.27 [0.63;2.54] | 3.02[1.10;8.33] | * | * | 1.80 [0.40;8.13] | 1.04 [0.24;4.51] | 0.90 [0.21;3.85] | 0.71 [0.16;2.98] |
| | 2 Eye | 1.33 [0.66;2.67] | 3.77[1.38;10.26] | * | * | 4.91 [0.90;26.83] | 2.91 [0.58;14.69] | 1.80 [0.38;8.48] | 1.03 [0.24;4.48] |
| | 3 Occupation | | 2.13[0.88;5.14] | * | * | * | * | * | * |
| Dementia | 1 Dislocation/sprain | 0.97 [0.59;1.60] | 0.73[0.29;1.84] | 1.96 [0.23;16.82] | 0.61 [0.08;4.62] | 0.29 [0.04;2.10] | 0.40 [0.10;1.64] | 0.59 [0.21;1.63] | 0.49 [0.18;1.35] |
| | 2 Eye | 1.15 [0.68;1.93] | 0.71[0.26;2.00] | 2.31 [0.26;20.78] | 0.86 [0.11;6.69] | 0.60 [0.08;4.52] | 0.88 [0.21;3.75] | 1.26 [0.44;3.58] | 1.09 [0.38;3.07] |
| | 3 Occupation | | 0.81[0.32;2.03] | * | * | * | * | * | * |
| Mental disorders due to known physiological condition | 1 Dislocation/sprain | 1.77 [1.22;2.56] | 1.99[1.24;3.17] | 4.44 [1.37;14.43] | 3.50 [1.38;8.87] | 3.16 [1.69;5.89] | 2.41 [1.34;4.31] | 1.84 [1.04;3.25] | 1.79 [1.05;3.06] |
| | 2 Eye | 2.06 [1.41;3.02] | 2.25[1.38;3.65] | 2.00 [0.58;6.90] | 2.50 [0.94;6.66] | 3.66 [1.88;7.12] | 2.46 [1.34;4.51] | 1.78 [0.99;3.22] | 1.88 [1.08;3.27] |
| | 3 Occupation | | 4.22[2.57;6.95] | 4.00 [0.78;20.61] | 6.67 [1.88;23.62] | 10.00 [4.16;24.03] | 9.17 [4.04;20.77] | 6.11 [2.89;12.94] | 5.22 [2.60;10.49] |
| Depression | 1 Dislocation/sprain | 1.11 [0.92;1.33] | 1.32[1.06;1.65] | 1.87 [0.98;3.56] | 1.50 [0.91;2.46] | 1.15 [0.78;1.70] | 1.14 [0.83;1.58] | 1.23 [0.94;1.62] | 1.18 [0.91;1.52] |
| | 2 Eye | 1.19 [0.98;1.43] | 1.32[1.05;1.66] | 2.12 [1.07;4.20] | 1.52 [0.88;2.62] | 1.04 [0.69;1.57] | 1.16 [0.83;1.61] | 1.31 [0.99;1.74] | 1.23 [0.95;1.61] |
| | 3 Occupation | | 1.71[1.37;2.13] | 3.33 [1.32;8.40] | 2.56 [1.27;5.13] | 1.61 [0.91;2.82] | 1.78 [1.15;2.76] | 1.95 [1.37;2.79] | 1.87 [1.34;2.60] |
| Anxiety and adjustment disorders | 1 Dislocation/sprain | 1.43 [1.24;1.66] | 1.66[1.39;1.97] | 3.65 [2.28;5.82] | 2.72 [1.93;3.83] | 2.06 [1.58;2.69] | 1.82 [1.45;2.28] | 1.72 [1.41;2.11] | 1.58 [1.30;1.91] |
| | 2 Eye | 1.51 [1.30;1.75] | 1.57[1.31;1.87] | 2.85 [1.81;4.50] | 2.52 [1.79;3.56] | 2.04 [1.56;2.68] | 1.91 [1.51;2.42] | 1.79 [1.46;2.21] | 1.73 [1.42;2.10] |
| | 3 Occupation | | 2.08[1.75;2.47]^ | 5.16 [2.81;9.44] | 4.82 [3.10;7.51] | 3.40 [4.41;4.79] | 2.93 [2.19;3.92] | 2.67 [2.05;3.49] | 2.44 [1.90;3.13] |
| Post Traumatic Stress Disorder (PTSD) | 1 Dislocation/sprain | 1.61 [1.02;2.56]^ | 1.83[1.04;3.22]^ | 13.33 [2.98;59.57] | 10.00 [3.23;31.01] | 4.71 [2.03;10.90] | 3.43 [1.78;6.60] | 2.88 [1.62;5.12] | 2.79 [1.63;4.77] |
| | 2 Eye | 1.67 [1.03;2.71]^ | 1.39[0.75;2.58] | 10.00 [2.50;39.98] | 5.56 [1.86;16.58] | 2.40 [0.98;5.85] | 3.33 [1.63;6.82] | 2.95 [1.59;5.49] | 2.68 [1.52;4.74] |
| | 3 Occupation | | 2.98[1.64;5.41]^ | * | * | 15.00 [4.23;53.15] | 7.50 [3.16;17.80] | 6.00 [2.63;13.71] | 6.11 [2.88;12.94] |
| Somatoform disorders | 1 Dislocation/sprain | 1.37 [0.87;2.16] | 1.76[0.97;3.17] | 1.41 [0.17;11.46] | 1.99 [0.56;6.87] | 3.20 [1.51;6.80] | 2.34 [1.16;4.30] | 1.99 [1.10;3.61] | 1.87 [1.08;3.24] |
| | 2 Eye | 1.36 [0.86;2.18] | 1.38[0.76;2.49] | 0.61 [0.08;4.59] | 1.18 [0.36;3.91] | 2.34 [1.13;4.84] | 2.11 [1.06;4.17] | 1.70 [0.92;3.14] | 1.78 [1.00;3.15] |
| | 3 Occupation | | 2.50[1.39;4.48] | * | 4.00 [0.78;20.62] | 2.86 [0.94;8.68] | 2.50 [0.94;6.66] | 2.30 [0.95;5.59] | 2.30 [0.95;5.59] |
| Other non-psychotic mental disorders | 1 Dislocation/sprain | 1.94 [0.87;4.37]^ | 1.18[0.27;5.26] | * | * | * | 0.91 [0.12;7.04] | 1.11 [0.26;4.79] | 0.83 [0.20;3.53] |
| | 2 Eye | 2.41 [1.06;5.51]^ | 2.07[0.44;9.76] | * | * | * | 0.63 [0.08;4.71] | 1.18 [0.27;5.09] | 1.11 [0.26;4.79] |
| | 3 Occupation | | 1.33[0.30;5.83] | * | * | * | * | 2.00 [0.23;17.12] | 1.43 [0.18;11.61] |

(*Continued*)

**Table 2.** (Continued)

| Outcome | Match | Time to event HR | Time to event Workforce only HR | 6 months OR | 12 months OR | 2 years OR | 3 years OR | 4 years OR | 5 years OR |
|---|---|---|---|---|---|---|---|---|---|
| Sleep disorders | 1 Dislocation/sprain | 1.40 [0.70;2.82] | 2.21[1.00;4.89] | * | * | 0.59 [0.08;4.42] | 1.20 [0.36;3.98] | 0.97 [0.30;3.17] | 1.14 [0.41;3.22] |
| | 2 Eye | 1.76 [0.87;3.58] | 2.24[1.02;4.92] | * | * | 1.11 [0.14;8.77] | 2.00 [0.58;6.91] | 1.58 [0.47;5.34] | 1.54 [0.54;4.41] |
| | 3 Occupation | | 2.96[1.34;6.52] | * | * | 1.42 [0.18;11.61] | 4.29 [1.11;16.57] | 2.31 [0.66;8.10] | 2.31 [0.66;8.10] |
| Sexual dysfunction | 1 Dislocation/sprain | 1.08 [0.68;1.74] | 1.21[0.73;2.03] | * | 0.53 [0.07;3.93] | 1.00 [0.36;2.79] | 0.78 [0.31;1.94] | 0.90 [0.41;1.94] | 0.85 [0.41;1.75] |
| | 2 Eye | 0.95 [0.58;1.56] | 0.91[0.53;1.55] | * | 0.36 [0.05;2.61] | 0.63 [0.23;1.74] | 0.62 [0.25;1.52] | 0.74 [0.34;1.59] | 0.60 [0.28;1.28] |
| | 3 Occupation | | 1.55[0.93;2.58] | * | 0.77 [0.10;5.88] | 1.43 [0.50;4.07] | 1.14 [0.41;3.22] | 1.30 [0.56;3.05] | 1.23 [0.56;2.69] |

HR: Hazard Ratio.

OR: Odds Ratio.

* Too few events to estimate risk.

^Proportional hazard not present.

'mental disorders due to known physiological condition', 'anxiety and adjustment disorders', especially the 'Post Traumatic Stress Disorder (PTSD)' subgroup. 'Depression' showed weaker association and only shortly after the injury, while others took time to develop (somatoform

**Table 3.** Sensitivity analyses (electrical injuries matched with dislocation/sprain controls).

| Outcome | Time to event from Table 3, match 1 HR | Hospitalization >1 day HR | Concussion excluded HR | Men HR | Women HR |
|---|---|---|---|---|---|
| Alzheimer's dementia | 1.27[0.63;2.54] | 1.10[0.33;3.62] | 1.19[0.59;2.37] | 1.13 [0.45;2.84] | 1.27 [0.45;3.61] |
| Dementia | 0.97[0.59;1.60] | 0.64[0.26;1.59] | 0.92[0.56;1.52] | 1.11 [0.64;1.93] | 0.52 [0.16;1.65] |
| Mental disorders due to known physiological condition | 1.77[1.22;2.56] | 2.31[1.23;4.32] | 1.72[1.18;2.47] | 1.32 [0.83;2.11] | 3.08 [1.65;5.73] |
| Mood affective disorders | 1.09[0.92;1.30] | 1.40[1.05;1.87] | - | 1.21 [0.97;1.50] | 0.91 [0.69;1.20] |
| Depression | 1.11[0.92;1.33] | 1.41[1.04;1.92] | - | 1.21 [0.96;1.53] | 0.97 [0.73;1.29] |
| Anxiety and adjustment disorders | 1.43[1.24;1.66] | 1.59[1.22;2.08] | - | 1.40 [1.16;1.68] | 1.48 [1.20;1.84] |
| Post Traumatic Stress Disorder (PTSD) | 1.61[1.02;2.56] | 1.67[0.65;4.30] | - | 1.40 [0.78;2.50] | 1.99 [0.93;4.25] |
| Somatoform disorders | 1.37[0.87;2.16] | 1.40[0.60;3.28] | - | 1.68 [0.99;2.86] | 0.78 [0.31;1.94] |
| Other non-psychotic mental disorders | 1.94[0.87;4.37] | 1.67[0.20;13.84] | - | 2.27 [0.86;6.00] | 1.43 [0.32;6.29] |
| Symptoms and signs involving emotional state | 1.54[0.84;2.83] | 3.00[0.83;10.90] | - | 1.09 [0.43;2.74] | 2.81 [1.34;5.89] |
| Sleep disorders | 1.40[0.70;2.82] | 2.94[1.09;7.97] | - | 1.25 [0.53;2.91] | 1.88 [0.55;6.43] |
| Sexual dysfunction | 1.08[0.68;1.74] | 1.00[0.40;2.51] | - | 1.07 [0.61;1.90] | 1.08 [0.47;2.53] |

disorders), or were only present in the time to event analysis ('other non-psychotic mental disorders' and 'sleep disorders'). Although the relative risk of some of the outcomes was high, this does not necessarily reflect a high absolute risk. Many of the outcomes were rare, despite a large amount of exposed persons.

Restricting the analysis to persons from the workforce generally attenuated the estimates slightly in match 1 and 2 but with decreased precisions due to limited power. In match 3 with only persons from the workforce and not controls with another accident, the estimates were also higher compared to match 1 and 2. The gender differences in the exposure to electrical injuries are well-know, but the differences in risk for men and women in mental disorders due to known physiological condition and somatoform disorders point in different directions and further research in this topic is needed.

A limitation of the study was the lack of registration in DNPR in the early years of the study. Probably only a low proportion of electrical injuries were registered, especially during the first years of the study. If the types of electrical injuries that were registered differed in type, severity or duration from those not registered, this may cause bias in an unknown direction. Even today, underreporting may be a problem, since the detailed registration may be deprioritized in acute situations. If the electrical injuries not registered in DNPR are the more serious injuries, where the consequence of the injury, i.e. a burn, was registered instead, and not the code for the electrical injury itself, we may have overlooked the most severe injuries, and thus underestimated the associations. Misclassification of the outcomes is possible in both the electrical injured persons and their match persons. Especially the rare outcomes are vulnerable to misclassification, where only one more person with the outcome in either group could impact the risk estimates to a large extent. Some of the outcomes may be prone to risk of misclassification as the exposure may be interpreted as part of the outcome. This is relevant for the PTSD outcome, but also for 'somatoform disorders' and 'mental disorders due to known physiological condition' where physical or cognitive symptoms are reported from the person as a consequence of an electrical injury and other diagnoses are not relevant. In the part of the cohort derived from DWEA, we have no reason to think that there is any difference in the reporting of the exposure, although the number of injuries decreased over time, which is also the case in other types of working injuries. The severity and other characteristics of the electrical injury were not registered, since the definition of an electrical injury was based on the ICD-10 code in DNPR and type of injury in DWEA. Other previous studies have distinguished between high and low voltage injuries. Radulovic et al. found no differences in neuropsychological sequelae, although high voltage injuries causes more physical damage [7], while Rådman et al. found higher incidence of sleep disturbances, anxiety and fatigue in high voltage injuries and of concentration difficulties, sleep disturbances and anxiety in 'no-let-go' injuries [36]. We tried to accommodate this by restricting the analysis to persons with a length of hospitalization of at least one day as a proxy for severity, and found that most estimates attenuated, meaning that the risk of mental diseases were higher in those persons that were most affected by the injury.

The hospitalization was in most cases very short. A large proportion of electrical injuries registered in DWEA either did not cause hospitalization or caused only an outpatient visit. This is surprising because the definition of an occupational accident that should be reported to DWEA is an accident that has caused at least sick-leave the day after the accident, and thus of a certain severity. On the other hand, it is possible that some persons were seen by their general practitioner, and thus not registered in DNPR.

Another limitation was the choice of match persons. We found it difficult to identify the perfect type of injury to match an electrical injury. Since electrical injuries are heterogeneous in severity, we should optimally use a similar heterogenetic group and at the same time it

should be a rather frequent type of injury to be able to find a sufficient amount of suitable match persons. Furthermore, we should use only matching diagnoses that are not suspected to cause the outcomes of interest. Our solution was to use three different types of match persons: persons with dislocation/sprain injury, persons with eye injury and finally persons with the same type of work as the injured persons. The first two matches had the disadvantage that the injuries were not in themselves life-threatening or disabling as an electrical injury can be. This would cause us to overestimate the frequency of our outcomes, especially regarding PTSD since been exposed to a traumatic event is per definition part of the outcome. On the other hand, those match persons shared the characteristic that they also visited a hospital for an injury the same year and thus shared the same health care seeking preferences as the injured persons.

Match 3 using persons with the same type of job had the disadvantage that the match persons did not have any registered injury, and thus were most likely not in the health care system at the time of the match. This implies that the estimates in this match are most likely overestimated, if injured persons had other habits in relation to health care seeking. In this case, we could not solve this possible bias by adjustment for length of hospitalization. The same approach was used in another Danish cohort study that matched electrical injured persons with random controls from the general population using age and gender with cardiac diseases and mortality as outcomes [37]. We aimed to match with other injured persons (match 1 and 2) to avoid using too healthy controls, but also to take socioeconomic position into account when matching with occupation controls (match 3).

The size of the study was the largest possible using Danish data. Even though DNPR was established in 1977, information about electrical injuries was not sufficiently registered before the initiation of ICD-10 in 1994. Since we chose to use two years of observation time before the accident to exclude persons with the outcome of interest and two years of observation time for new outcomes to occur, we were limited to 19 years from 1996 to 2014. Despite the long period and the considerable amount of injuries, we still had problems with limited power in some analyses, especially with rare diagnoses and when the outcome had occurred before the injury. The latter could cause bias if the outcome was related to a previous unregistered electrical injury. However, since the matching was performed using the injury year, both electrical injured persons and controls were at the same risk of overlooking previous exposure and outcome and thus the bias in most likely non-differential.

Our findings were in line with several previous studies with more retrospective or casuistic design, and the reviews concluding on these. We were not able to compare the size of the risk estimates to previous studies, since the control groups and outcomes from previous studies are not comparable to our design. However, we were able to confirm the findings of both cognitive and psychological problems described in previous studies [4, 8, 11, 22].

The findings related to Alzheimer's Dementia were surprising, especially that increased risk was found when restricting the analysis to the working population. A possible explanation is that the disease was already in development at the time of the injury and other examinations following the injury caused the disease to be diagnosed before it would have been diagnosed without the health contact caused by the injury. On the other hand, only very few diagnoses were given during the first 5 years following the injury, and thus not related to health contacts following the injury.

Some of the chosen outcomes could be related to the electrical injury itself, i.e. PTSD where the diagnosis is based on a specific traumatic event and 'mental disorders due to known physiological condition' could possibly be a diagnosis coded as the physician knows about the exposure to an electrical injury and thus codes cognitive problems according to this.

The findings from this study can be generalized to populations with the same access to hospital treatment and/or a similar system for registration of working injuries. Moreover, the excess risk for mental disorders is most likely also present in other kinds of populations, due to the matched design that reduce confounding [38]. As already suggested by Radulovic et al, more focus on this topic as well as on rehabilitation is warranted [7]. A study in patients from an outpatient burn department found that patients were to a large extend referred to other specialist departments, but often with no clear conclusion or diagnosis [20]. This was in line with our finding from this cohort study, were we found that electrical injures patients had more contact with general practitioners as well as more sick leave and poorer work participation [39].

## Conclusion

This study confirms that exposure to an electrical injury increase the risk of mental disorders both at long and short term. Some of the outcomes were rare, thus the high relative risk is not reflected in the absolute risk which is low. To our knowledge, this is the first study to examine mental disorders in a controlled design.

## Author Contributions

**Conceptualization:** Karin Biering, Jesper Medom Vestergaard, Anette Kærgaard, Ole Carstensen, Kent J. Nielsen.

**Data curation:** Karin Biering, Jesper Medom Vestergaard.

**Formal analysis:** Karin Biering.

**Funding acquisition:** Kent J. Nielsen.

**Investigation:** Karin Biering, Jesper Medom Vestergaard.

**Methodology:** Karin Biering, Jesper Medom Vestergaard, Anette Kærgaard.

**Project administration:** Karin Biering, Kent J. Nielsen.

**Supervision:** Kent J. Nielsen.

**Writing – original draft:** Karin Biering.

**Writing – review & editing:** Karin Biering, Jesper Medom Vestergaard, Anette Kærgaard, Ole Carstensen, Kent J. Nielsen.

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
