## [Decision Letter · Decision Letter 0]

30 Nov 2020

PONE-D-20-30102

Mental disorders following electrical injuries – a register-based, matched cohort study

PLOS ONE

Dear Dr. Biering,

Thank you for submitting your manuscript to PLOS ONE. After careful consideration, we feel that it has merit but does not fully meet PLOS ONE’s publication criteria as it currently stands. Therefore, we invite you to submit a revised version of the manuscript that addresses the points raised during the review process, which you find below.

In particular, you should address the selection of your control group as noted by Reviewer 2. Do you mean "strain trauma" by "distortion"?

We look forward to receiving your revised manuscript.

Kind regards,

Thomas Behrens

Academic Editor

PLOS ONE

3.We note that you have indicated that data from this study are available upon request. PLOS only allows data to be available upon request if there are legal or ethical restrictions on sharing data publicly. For information on unacceptable data access restrictions, please see http://journals.plos.org/plosone/s/data-availability#loc-unacceptable-data-access-restrictions.

4. We noted in your submission details that a portion of your manuscript may have been presented or published elsewhere.

[Yes, Figure 1 the flowchart and Table 1 appears in a paper under review elsewhere]

5. Please provide additional details regarding participant consent. In the Methods section, please ensure that you have specified (1) whether consent was informed and (2) what type you obtained (for instance, written or verbal). If your study included minors, state whether you obtained consent from parents or guardians. If the need for consent was waived by the ethics committee, please include this information.

**Comments to the Author**

1. Is the manuscript technically sound, and do the data support the conclusions?

Reviewer #1: No

Reviewer #2: Yes

2. Has the statistical analysis been performed appropriately and rigorously? 

Reviewer #1: Yes

Reviewer #2: I Don't Know

3. Have the authors made all data underlying the findings in their manuscript fully available?

Reviewer #1: Yes

Reviewer #2: Yes

4. Is the manuscript presented in an intelligible fashion and written in standard English?

Reviewer #1: Yes

Reviewer #2: Yes

5. Review Comments to the Author

Reviewer #1: I think this is an important study but I do have concerns.

why did you not chose burn injuries with small burns as your controls? eye injuries and distorsion are very different and do not lead to PTSD in general i find that you need to control vs thermal injuries.

any data on low voltage vs high voltage?

any data on the support system? did the patients have access to mental health or peer groups?

what is the incidence of pre-existing mental alterations? others showed that EI with mental health issues have had pre-existing ones.

Others showed MDD and Anxiety. what is your finding?

it is not the first study linking EI to mental health. ffish, gomez, jeschke all did.

plse edit MS to more focused and add hypothesis and aim.

Reviewer #2: This careful march through a connected web of datasets provides an important piece of data on electrical injury outcomes. A few comments on the manuscript are listed below:

1. Most clinicians are not coders and won't know immediately or at all what a "distortion" refers to. Recommend describing this further in clinical terms.

2. A significant portion of the most seriously injured electrical injury survivors would be expected to have pain, major physical deficits or visible differences such as burn scars, contractures, paralysis, amputation, blindness, deafness, or heterotopic ossification which by themselves might contribute to poor psychological outcomes. More literature references differentiating electrical injury survivors from flame and other etiology burn survivors would support the conclusions of the authors. Why did the authors not choose to have a non-electrical burn control group as well?

3. The authors should discuss possible reasons why electrical injury survivors would/could have mental health issues long-term?

4. How are these conclusions different from patient reported outcomes data on electrical injuries?

5. What are the next steps?

6. PLOS authors have the option to publish the peer review history of their article (what does this mean?). If published, this will include your full peer review and any attached files.

Reviewer #1: No

Reviewer #2: No

---

## [Author Response · Author response to Decision Letter 0]

20 Jan 2021

Authors reply: The data are situated on Statistic Denmark's secure servers for researchers. Access to the data can be obtained by application to Statistics Denmark with some limitations: https://www.dst.dk/en/TilSalg/Forskningsservice

Authors reply: See above

4. We noted in your submission details that a portion of your manuscript may have been presented or published elsewhere.

[Yes, Figure 1 the flowchart and Table 1 appears in a paper under review elsewhere]

Authors reply: The flowchart (Figure 1) and table 1 describes the data collection, matching and demographics of the study population and since the same material was used in another publication (with completely different focus), we think that the figure and the table is necessary in both publications. None of the result in the paper can be derived from these, as they are purely descriptive.

Update january 2021: the Figure and the table is now removed from the manuscript, ands replaced with a link to the paper, according to guidelines from the editorial office.

5. Please provide additional details regarding participant consent. In the Methods section, please ensure that you have specified (1) whether consent was informed and (2) what type you obtained (for instance, written or verbal). If your study included minors, state whether you obtained consent from parents or guardians. If the need for consent was waived by the ethics committee, please include this information.

Authors reply: All data are register based and thus participants consent is not possible and not necessary according to Danish Law, See: https://www.retsinformation.dk/eli/lta/2017/1083 (in Danish, §10)

Comments to the Author

1. Is the manuscript technically sound, and do the data support the conclusions?

Reviewer #1: No

Authors reply: We hope that the responses explained in the following will alter this perception.

Reviewer #2: Yes

2. Has the statistical analysis been performed appropriately and rigorously? 

Reviewer #1: Yes

Reviewer #2: I Don't Know

3. Have the authors made all data underlying the findings in their manuscript fully available?

Reviewer #1: Yes

Reviewer #2: Yes

4. Is the manuscript presented in an intelligible fashion and written in standard English?

Reviewer #1: Yes

Reviewer #2: Yes

5. Review Comments to the Author

Reviewer #1: I think this is an important study but I do have concerns.

why did you not chose burn injuries with small burns as your controls? eye injuries and distorsion are very different and do not lead to PTSD in general i find that you need to control vs thermal injuries.

Authors reply: Thank you for this question. The choice of match persons was something that we discussed several times in the author group. The idea behind matching in cohort studies, is to choose controls that are unexposed, compared to the group of interest (Modern Epidemiology (3rd edition), Rothmann et al. 2008), but we used other injured persons in the two first matches, to avoid that the exposure of having any kind of injury would affect the outcome in itself and not just the exposure to electrical injury. We furthermore included matching with persons from the same occupation group, to take socio-demographic differences into account, although these were not injured controls. We did not choose other burn patients as controls, since this was a small group with limited possibilities to find sufficient match persons, but more importantly, since the registration of electrical injuries were most likely insufficient in the first years of the study, we were worried that electrical injuries were registered only with burn diagnosis, and not with an electrical injury. This could imply that the exposure was not different in the electrical injuries and the match persons as required. We have added a sentence regarding PTSD in the discussion section.

any data on low voltage vs high voltage?

Authors reply: Unfortunately, we did not have information about voltage or other characteristics of the electrical exposure, since the register included only ICD-10 coding. However, we are currently working on a cohort study, that may meet some of these limitations, see details in answer to reviewer #2.

any data on the support system? did the patients have access to mental health or peer groups?

Authors reply: We do not have information regarding possible support systems in general, but the Danish public health system provides the same access to health services based on indication and thus possibilities for support. In a previous study we found that electrical injured patients had more contact with general practitioner, compared to the match persons, maybe reflecting a need for support. (Biering et al. 2020)

what is the incidence of pre-existing mental alterations? others showed that EI with mental health issues have had pre-existing ones.

Authors reply: We did have information regarding mental diseases before the EI, if these were diagnosed in hospital settings (in- and outpatients). In table 2, we have provided the numbers of IE with pre-existing mental conditions, and these were excluded in the analysis.

Others showed MDD and Anxiety. what is your finding?

Authors reply: We based our findings on depression on the two ICD-10 groups DF32 and DF33 and 'anxiety and adjustment disorders' on the groups DF40, DF41, DF42, DF43 and DF44.

 We found that these two groups provided the most frequent outcomes compared to other mental health outcomes, and with increased risk compared to the match-groups. The risk was highest shortly after the IE. (see table 3 for details)

it is not the first study linking EI to mental health. ffish, gomez, jeschke all did.

plse edit MS to more focused and add hypothesis and aim.

Authors reply: We agree that this is indeed not the first study to link electrical injuries to mental health problems, but to our best knowledge, it is the first study to use registerbased information to include comparison groups without electrical injuries. We have edited the introduction section to include the Fish reference as well as a recent Swedish publication, but based om this limited information, we were not able to identify the two other papers in our systematic search or in a free-text search using Author name and Electrical injuries in combination. We have changed the wording in the introduction to include hypothesis. Furthermore, we have included a few more specific references for the registries used.

Reviewer #2: This careful march through a connected web of datasets provides an important piece of data on electrical injury outcomes. A few comments on the manuscript are listed below:

1. Most clinicians are not coders and won't know immediately or at all what a "distortion" refers to. Recommend describing this further in clinical terms.

Authors reply: Thank you for the kind words regarding our study. We have had the same feedback regarding the wording of distortion from others colleagues and have consequently changed this throughout the manuscript to 'dislocation or sprain'

2. A significant portion of the most seriously injured electrical injury survivors would be expected to have pain, major physical deficits or visible differences such as burn scars, contractures, paralysis, amputation, blindness, deafness, or heterotopic ossification which by themselves might contribute to poor psychological outcomes. More literature references differentiating electrical injury survivors from flame and other etiology burn survivors would support the conclusions of the authors. Why did the authors not choose to have a non-electrical burn control group as well?

Authors reply: Since the majority (80%) of our population of electrical injured persons had a very short hospital contact, our population was most likely not as severely injured as patients recruited from burn units. See also reply to reviewer #1 regarding methodology consideration in the matched design.

3. The authors should discuss possible reasons why electrical injury survivors would/could have mental health issues long-term?

Authors reply: It is difficult to state exact reasons for this, but we can guess that it is a range of circumstances, first of all the immediate shock of having been in a potential life threatening situation. Furthermore, we know from electrical injured patients that we meet here in our clinical setting of occupational medicine, that many patients suffer from unspecific symptoms and complaints, that are not understood by employers, relatives or even the health care system. Frustrations and insecurity regarding health and rehabilitation may affect the mental health. We have added this to the discussion section.

4. How are these conclusions different from patient reported outcomes data on electrical injuries?

Authors reply: First of all, this study includes a variety of electrical injuries, many with a short contact with the hospital, and thus with less severity compared to several previous studies from burn units. Furthermore, we were able to use the matched design, to have a comparison group for the analyses.

5. What are the next steps?

Authors reply: We are thrilled that you ask, because while we have conducted this register-based study, we have also collected data for a cohort study with 6960 members from the Danish Union of Electricians during 6 months with weekly data on health and exposure to electrical shocks in detail, in order to meet some of the limitations in this study, especially the lack of details regarding exposure.

6. PLOS authors have the option to publish the peer review history of their article (what does this mean?). If published, this will include your full peer review and any attached files.

Do you want your identity to be public for this peer review? For information about this choice, including consent withdrawal, please see our Privacy Policy.

Reviewer #1: No

Reviewer #2: No

---

## [Decision Letter · Decision Letter 1]

5 Feb 2021

Mental disorders following electrical injuries – a register-based, matched cohort study

PONE-D-20-30102R1

Dear Dr. Biering,

We’re pleased to inform you that your manuscript has been judged scientifically suitable for publication and will be formally accepted for publication once it meets all outstanding technical requirements.

Kind regards,

Thomas Behrens

Academic Editor

PLOS ONE

Additional Editor Comments (optional):

Reviewers' comments:

Reviewer's Responses to Questions

**Comments to the Author**

1. If the authors have adequately addressed your comments raised in a previous round of review and you feel that this manuscript is now acceptable for publication, you may indicate that here to bypass the “Comments to the Author” section, enter your conflict of interest statement in the “Confidential to Editor” section, and submit your "Accept" recommendation.

Reviewer #1: All comments have been addressed

Reviewer #2: All comments have been addressed

2. Is the manuscript technically sound, and do the data support the conclusions?

Reviewer #1: (No Response)

Reviewer #2: Yes

3. Has the statistical analysis been performed appropriately and rigorously? 

Reviewer #1: (No Response)

Reviewer #2: Yes

4. Have the authors made all data underlying the findings in their manuscript fully available?

Reviewer #1: (No Response)

Reviewer #2: Yes

5. Is the manuscript presented in an intelligible fashion and written in standard English?

Reviewer #1: (No Response)

Reviewer #2: Yes

6. Review Comments to the Author

Reviewer #1: no further comments needed or required

xxxxxxxxxxxxxxxxxxxxxxxxxxxxxxxxxxxxxxxxxxxxxxxxxxxxxxxxxxxxxxxxxxxxxxxxxxxxxx

Reviewer #2: (No Response)

7. PLOS authors have the option to publish the peer review history of their article (what does this mean?). If published, this will include your full peer review and any attached files.

Reviewer #1: No

Reviewer #2: No

---

## [Editor Report · Acceptance letter]

9 Feb 2021

PONE-D-20-30102R1 

Mental disorders following electrical injuries – a register-based, matched cohort study 

Dear Dr. Biering:

I'm pleased to inform you that your manuscript has been deemed suitable for publication in PLOS ONE. Congratulations! Your manuscript is now with our production department. 

Kind regards, 

on behalf of

Prof. Thomas Behrens 

Academic Editor

PLOS ONE